# Parenting Practice Profiling and Its Associated Factors among Secondary Vocational School Students in China

**DOI:** 10.3390/ijerph19127497

**Published:** 2022-06-18

**Authors:** Yujia Zheng, Yuhang Fang, Yan Jin, Xiayun Zuo, Qiguo Lian, Chaohua Lou, Chunyan Yu, Xiaowen Tu, Lihe Li, Ping Hong

**Affiliations:** 1NHC Key Laboratory of Reproduction Regulation (Shanghai Institute for Biomedical and Pharmaceutical Technologies), Fudan University, Shanghai 200237, China; 19211150005@fudan.edu.cn (Y.Z.); fangyuhang@sibpt.com (Y.F.); zuoxiayun@sibpt.com (X.Z.); lianqiguo@sibpt.com (Q.L.); louchaohua@sibpt.com (C.L.); yuchunyan@sibpt.com (C.Y.); 2Shaanxi Xin Hang Public Health Research Center, Xi’an 710061, China; jinyan@xinhangxinyu.com; 3China Family Planning Association, Beijing 100035, China; lihecfpa@163.com (L.L.); hongpingcfpa@163.com (P.H.)

**Keywords:** parenting style, latent profile analysis, secondary vocational school, students, China

## Abstract

Background: Parenting styles have a tremendous influence on a child’s development and behavior. Studies on parenting styles using latent profile analysis have been increasing in recent years. However, there are few such studies conducted in China, especially concerning joint parenting styles (that simultaneously characterize maternal and paternal practices), which are held over the age group of secondary vocational school students. This study aimed to identify the profiles of parenting styles and their associated factors among the parents of secondary vocational school students in China, based on natural samples and not a predetermined model. Method: Data were drawn from a cross-sectional study conducted among 3180 students from six secondary vocational schools in Shanghai Municipality and Shaanxi Province. A total of 2392 students who have lived with their parents for most of their lifetime were included in the study. Latent profile analysis was used to identify the profiles of parenting styles of the respondent’s parents. Multinominal logistic regression models were used to examine the association between parenting style and demographic characteristics and family background and adolescent outcomes. Results: We identified five latent profiles: “free-range parenting” (27.05%), “behavioral monitoring parenting” (33.65%), “authoritative parenting” (11.75%), “psychological control parenting” (14.38%) and “tiger parenting” (13.17%). The associations between these profiles and adolescent outcomes indicated that these profiles were rational. Participants’ gender, grade, residential area, family economic level, parental marital relationship, and parental educational level were predictive factors for parenting styles. Conclusions: The parenting styles held over secondary vocational school students were somewhat different from Baumrind’s parenting style model. A considerable number of students received a parenting style that might predispose them to behavioral and mental health outcomes and merit a tailored intervention using the predictive factors of parenting styles.

## 1. Introduction

A large number of studies have documented that parenting styles have a tremendous influence on the development of children, such as lifestyles [1], behavior problem [2], mental health [3], and well-being [4]. Parenting style, which means a steady pattern or inclination of parenting behavior, was first described by Baumrind (1967) [5]. Baumrind (1967) and Maccoby and Martin (1983) proposed a parenting style model with four parenting styles based on two broad dimensions of responsiveness (the extent to which parents intentionally foster the children’s individuality, self-regulation, and self-assertion by being attuned, supportive, and acquiescent to the children’s special needs and demands) and demandingness (the pressure parents put on children to become integrated into the family whole by maturity demands, supervision, disciplinary efforts, and a willingness to confront a child who disobeys) [6]. The four parenting styles are: authoritative parenting (high responsiveness and high demandingness), authoritarian parenting (low responsiveness and high demandingness), permissive parenting (high responsiveness and low demandingness), and neglecting parenting (low responsiveness and low demandingness). This model has been widely used.

However, despite there being many studies on parenting styles, most of them actually were conducted via a variable-centered approach instead of as a whole parenting style. Although a variable-centered approach can exploit all available data and evaluate the effects of each parenting dimension independently [7], it is inappropriate for studies of parenting. For example, simple dimensions do not affect the children’s development independently but play a role together under the influence of the overall parenting style [8]. In addition, some researchers have argued that the basis of the classification of this model is simple and extreme by arbitrary cut-off scores, predetermined by researchers on two broad dimensions, which might leave out the parenting styles that are beyond the well-established model. Recent studies began to apply latent profile analysis in the field of parenting and found that there were other parenting styles beyond the four classical styles [9,10], such as controlling-indulgent parenting [11] and unlabeled parenting [12]. Researchers also noticed that the four classic parenting styles identified from western samples may not be applicable in China, where parents are deeply influenced by Confucianism, which emphasizes collectivism and filial piety through: social interdependence, respect for parents and elders, fulfillment of family obligations, and conforming and obedience to norms [13].

By using dimensions that include warmth, inductive reasoning, encouragement of independence, encouragement of achievement, supervision, and harness, Zhang et al. [10] found that in China, besides *authoritative* parenting and *authoritarian* parenting, which were already recognized, parents of early adolescents (from fifth grade to eighth grade) had two other parenting styles: *average-level undifferentiated* parenting (the score of no dimension was significantly higher or lower) and *high-level undifferentiated* parenting (the score of no dimension was significantly higher or lower, but the scores of all dimensions were relatively high). Contrary to the stereotype of Chinese parents preferring strictness and discipline, which are supposed to be beneficial to the development of children [13,14], the most popular parenting style found in Zhang’s study was *average-level undifferentiated* parenting, which comprised almost half of the sample, while authoritarian parenting only comprised 6.2–10.0%. This study challenged the classic parenting model and forced us to reconsider parenting styles in China. However, the study only focused on early adolescents and only measured maternal parenting. Paternal and maternal parenting both influence child development, for which each effect is dependent [15]. Although this is a recognized condition, only a few studies consider it.

Based on the scores of senior high school entrance examinations, students with lower scores will go to a secondary vocational school in China. According to the Statistical Communique of China on national education development (2019), enrollment in secondary vocational schools accounts for 41.7% of the total enrollment in post junior high school education [16]. Secondary vocational school students are a fragile yet neglected group with a higher prevalence of internalizing problems as well as having more problem behaviors than their peers in senior high schools [17]. In addition, compared to parents of senior high school students, parents of secondary vocational school students are characterized by higher rates of not in marriage (10.8% vs. 17.6%), lower educational levels (the proportion of mother and father with senior high school education are 69.6%, 71.7% vs. 52.0%, 57.0%, respectively), and a lower household monthly income (lower than 3000 CNY being 29.7% vs. 41.6%) [18]. Previous studies have suggested that parenting styles are associated with parental education level, marital status, family economic status, and the sex and age of the children [19]. It is necessary to understand the parenting styles of these parents and the corresponding influencing factors to inform the design of an intervention. However, there have been few studies that have focused on parenting styles among this group. The current study will measure both paternal and maternal parenting practices concurrently to identify the parenting profiles of secondary vocational school students.

The current study included two aims: (1) identifying the parenting profiles of secondary vocational school students’ parents, using latent profile analysis and examining their rationality with associations between parenting profiles and adolescent outcomes, and (2) exploring the associated factors of different parenting practices profiles.

## 2. Methods

### 2.1. Participants and Procedure

This was a cross-sectional study conducted in Shanghai municipality and Shaanxi Province from March through April 2021. Shanghai Municipality, a provincial administrative unit, is the most developed and open coastal city in Eastern China, with a permanent population of 24.87 million, and Shaanxi Province is a typically less-developed inland province with a permanent population of 39.53 million in Western China. We chose three secondary vocational schools from the central urban area, the outskirts, and the outer suburbs of Shanghai and three secondary vocational schools from regions with high, medium, and low economic levels in Shaanxi Province, respectively. Then we chose classes with an equal proportion of gender and grade by cluster sampling. A total of 3180 students participated, and 2392 students who lived with their parents for most of their lifetime were included in the current study. The study was approved by the ethics committee of the Shanghai Institute of Planned Parenthood Research. Parental consent and participants’ assent were collected before data collection.

### 2.2. Materials & Methods

#### 2.2.1. Parenting Practices

We measured parental–child connectedness, behavior monitoring, and psychological control as parenting dimensions for our study. Connectedness refers to a supportive and affectionate parent–child interaction. Behavior monitoring, which is an important dimension of behavioral control, refers to the monitoring of children’s activities and behaviors. Psychological control refers to parents trying to control the children’s thoughts or feelings through guilt induction, love withdrawal, shaming, and other disrespectful or manipulative behaviors, which will hinder the children’s development of identity and autonomy [20]. Behavioral control is linked with a series of positive outcomes, while psychological control is linked to negative outcomes, especially psychological problems, such as depression and anxiety [21]. Thus, many scholars argue that these two dimensions of control should be distinguished within research [20]. In general, the three dimensions we measured are similar to the two classic parenting dimensions of responsiveness and demandingness but are more distinguished by demandingness and also have consistency with the three critical components of parenting authority (involving behavioral monitoring and psychological autonomy) [22]. In addition, a recent study found that the parenting practices that adolescents perceived were more representative of their personal experiences and might be more predictive of their adjustments [23]. Thus, we measured parental practices from the perspective of adolescents. Both paternal and maternal parenting practices were measured.

***Parent–child connectedness*** was measured by a four-item scale developed by Sidze et al. [24]: (1) how often did your parents/guardians encourage you?, (2) how often were your parents/guardians interested in what you thought?, (3) how often did your parents/guardians try to find activities that you enjoyed?, and (4) how often did you talk to your parents/guardians when you had problems? Participants were asked these four questions using a four-point scale: 1 (never), 2 (sometimes), 3 (most of the time), and 4 (all the time).

***Behavioral monitoring*** was measured by the restriction scale of parents’ behavior control, developed by Wang et al. [25]. The questionnaire adopts a five-point scoring method from “never” to “always”. The higher the score, the stronger the parents’ behavior control over their children.

***Psychological control*** was measured by the following statements: my mother/father is a person who (1) is always trying to change how I feel or think about things, (2) brings up past mistakes when she/he criticizes me, and (3) is less friendly with me if I do not see things her/his way. Response options were five-point from “not like her/him at all” to “exactly like her/him” [26,27].

Cronbach’s alphas were 0.88 and 0.88 for paternal and maternal parent–child connectedness, 0.92 and 0.91 for paternal and maternal monitoring, 0.86 and 0.86 for paternal and maternal psychological control, respectively.

Individual scores were obtained by averaging the item scores for each dimension, with a higher score indicating a higher level of certain parenting dimension.

#### 2.2.2. Demographic Information

Demographic information included gender, grade, one child in the family or not, and resident area.

#### 2.2.3. Family Factors

Family factors included family economic level, parent’s marital relationship, and maternal and paternal educational level.

#### 2.2.4. Mental Health

Anxiety and depression were measured by the GAD-2 scale (α = 0.80) and PHQ-2 scale (α = 0.87), respectively, which have proven reliability and validity and could be used as screening tools [28,29,30]. We also asked participants whether they had ever attempted self-injury or suicide.

#### 2.2.5. Health Related Behaviors

“Attempting to smoke”, “attempting to drink” and “experience of sexual intercourse” were measured as health-related behaviors. Students answered with a “yes” or “no”.

#### 2.2.6. Academic Achievement

Students’ academic achievement was assessed by a self-evaluation questionnaire with a five-point answer scale, from “excellent” to “very poor”: How are your academic achievements in class? We reclassified these answers into “good”, “average” and “poor” during data analysis.

#### 2.2.7. Analytic Strategy

Latent profile analysis (LPA) was adopted to identify the profiles of parenting styles among respondents [31]. The best model was selected according to three aspects. Firstly, model selection started with a one-profile model and then the number of profiles increased until there was no significant model fit improvement or no addition of an interpretable profile. The best-fitting solution was determined by comparative model fitting, including Bayesian information criteria (BIC), Akaike information criterion (AIC), adjusted Bayesian Information criteria (aBIC), and Lo-Mendell-Rubin likelihood (LMR). Smaller BIC, aBIC, and AIC values indicate a better model fit. Entropy values indicate the accuracy of individual classifications. The larger the value, the better the corresponding model. When the *p* value of LMR and BLRT tests is less than 0.05, this indicates that the corresponding k-profile model is better than the model with k − 1 profiles. Secondly, the proportion of each profile shouldn’t be too small. Lastly, the interpretability of each profile was considered when choosing the best model. ANOVA was used to verify the newly generated parenting subtypes.

Then, multinominal logistic regression was employed to test the associations between the identified subtypes of parenting practice and adolescent outcomes after controlling for covariate variables.

Lastly, chi-square test and multinominal logistic regression were conducted to identify the factors associated with parenting styles.

LPA was conducted using Mplus 8.3 (Muthén & Muthén, Los Angeles, CA, USA). ANOVA, Chi-square test and multinominal logistic regression were conducted by Stata/SE 15.1 (StataCorp, College Station, TX, USA). We eliminated the observations which were absent from all items of any parenting practice scale (1.77%, 43/2435). Other missing data from parenting practices scales were filled in by average values (the observations that needed to be filled for each dimension was not more than 3.7%).

## 3. Results

### 3.1. Demographics

The mean age of our sample was 16.93 years (SD = 0.03). The sample was predominantly made up of students who: were girls (53.14%), from Eastern China (50.43%), had siblings (58.49%), and were from families with an average economic level (64.84%). As for the grade of the respondents, 35.62% were in grade one, 32.94% were in grade two, and 31.44% were in grade three. The marital relationships of their parents were mostly good (79.22%). Most of their fathers and mothers had senior higher school or lower education (81.67% vs. 83.72%, respectively).

### 3.2. Latent Profile Analysis

We conducted LPA modeling using the average scores of each scale (see Table 1 for the model fit statistics). A five-profile model was supported by fit indices, which had smaller AIC, BIC, and aBIC values and higher entropy values. The values for LMR and BLRT also showed statistical significance that five-profile model is better than the four-profile model. A six-profile model was also considered, however the added sixth profile was not easy to interpret and was similar to another profile from the same model. Also, the LMR of the six-profile model was close to 0.05, making it inappropriate to consider the six-profiles as being better than five-profiles. Therefore, we chose the five-profile model to represent the subtypes of parenting styles.

An ANOVA was used to verify the newly generated parenting subtypes. As shown in Table 2, the latent profile subtypes of parenting can accurately distinguish the degree of different parenting dimensions, indicating that this model was effective.

### 3.3. Profiles of Parenting Styles

The five-profile model is shown in Figure 1. *The first profile*, marked in light blue, comprised 27.05% of the sample. This group of parents showed moderate levels of parent–child connectedness and a low level of monitoring and psychological control over their children. This profile was called *free-range* parenting.

*The second profile,* marked in orange, comprised 33.65% of the sample. Parents in this group showed high levels of parent–child connectedness, a high level of monitoring and a low level of psychological control. This profile was called *behavioral monitoring* parenting.

*The third profile,* marked in gray, comprised 11.75% of the sample. Parents in this group showed a high level of parent–child connectedness and monitoring but the lowest level of psychological control. This profile was called *authoritative* parenting.

*The fourth profile,* marked in yellow, comprised 14.38% of the sample. This subtype of parenting practice showed low levels of parent–child connectedness and monitoring but a high level of psychological control. This profile was called *psychological control* parenting.

*The fifth profile,* marked in dark blue, comprised 13.17% of the sample. This subtype of parenting practice showed moderate levels of parent–child connectedness and the highest level of monitoring and psychological control, which makes this profile similar to the *tiger* parenting style. This profile was called *tiger* parenting.

### 3.4. The Associations between Profiles and Adolescent Outcomes

Table 3 shows that mental health, health-related behaviors, and academic achievement varied significantly among different parenting profiles. Adolescents with authoritative parents had the least negative outcomes and a higher academic achievement level, while adolescents with psychological control parents contrasted with this.

### 3.5. Influencing Factors of Parenting Styles

Table 4 and Table 5 show that respondents under free-range parenting tended to be in the advanced grades, live in Shanghai Municipality, and have a father with a higher education level. Respondents who reported authoritative parenting were more likely to be a girl, have a high family economic level, and have parents with a good marital relationship. Respondents who reported psychological control parenting were more likely to be a boy, within an advanced grade, have parents with a poor marital relationship, and have a mother with a higher education level. Additionally, parents who conducted tiger parenting on children were more likely to have a poor marital relationship. Both one-child families and paternal educational level were factors not associated with parenting profiles.

## 4. Discussion

To the best of our knowledge, this study identified the parenting profiles of secondary vocational school students by measuring both paternal and maternal parenting practices concurrently for the first time in China. Five parenting profiles were identified by LPA in our study: free-range, behavioral monitoring, authoritative, psychological control, and tiger subtypes. Although the profiles were different from Baumrind’s parenting style model, except for the authoritative parenting subtype, the associations between these profiles and adolescent outcomes were consistent with the effects of parenting practices on adolescent outcomes.

In traditional Chinese society, parents have the duty of teaching children and keeping their behavior in accordance with social norms, which means “to love”. In turn, children are expected to respect and obey their parents [32], which is called “Xiao” in Chinese. In our study, the fifth profile, called tiger parenting, conformed to the traditional impression on Chinese parents, such as having high levels of both authoritativeness and authoritarianism [33] or authoritative and psychological control [34]. This is similar to the tiger parenting profile which was first examined by Kim and his colleagues among adolescents aged 12 to 15 years old from Chinese American families, defined as having high levels of parental monitoring, psychological control, hostility, shaming, and punitive parenting, while having moderate to high levels of warmth, inductiveness, and democratic parenting [35]. A similar parenting style was also found in a study conducted among adolescents between fifth and eighth grades in the city of Jinan, Shangdong Province, in Eastern China [10]. However, contrary to traditional impressions, tiger parenting was not the main form of parenting style for secondary vocational school students in this study and for early adolescents in other studies in China [10]. Besides being challenged by western parenting culture since China’s reform and opening in the late 1970s, this may also be attributed to the influence of China’s one-child policy implemented from 1978 to 2015.

The common parenting styles in our study were free-range parenting (27.0%) and monitoring parenting (33.7%). The possible reason might be that secondary vocational school students are not pressured into entering a higher education school, unlike other students in general schools. Thus, their parents do not have high academic expectations for them and only require them to graduate normally. Moreover, secondary vocational school students are at the age of late adolescence, meaning some parents may only maintain a certain level of behavioral monitoring to ensure the safety of their children, and some parents may even completely relax the monitoring over their children at this age.

In addition, authoritative parenting accounted for only 11.7% of the sample (among parents of secondary vocational school students) in our study while accounting for 43.0% of the sample (early adolescents’ families) in Zhang’s study [10]. The parents in Zhang’s study having a much higher education level might have contributed to this difference. In Zhang’s study, the percentage of mothers and fathers with a college or higher education (58.9% and 67.8%, respectively) was significantly higher than those in our sample (16.3% and 18.3%, respectively). Previous studies show that a higher parental education level is associated with authoritative parenting [36]. Possibly due to the generally low education level of the parents, no significant association emerged between a parent’s education level and their parenting style in this study.

A noteworthy parenting style from our study was the psychological control parenting style, with a low level of parent–child connectedness and behavioral monitoring but a high level of psychological control, which had strong negative associations with adolescent outcomes and should be changed.

In accordance with previous studies, parenting styles from our study related to the respondent’s gender, grade, residential area, family economic level, and parental marital relationship. Students who were girls or from families with a high economic level or had parents with a good marital relationship were more likely to be reared by authoritative parenting, which is consistent with previous findings [37,38,39].

We noticed that the parenting subtype with the largest proportion was free-range parenting in Shanghai Municipality, while it was behavioral monitoring parenting in Shaanxi Province. A potential reason contributing to this significant regional difference could be that Shanghai Municipality is the most open and developed city in China, and people from Shanghai are more likely to be influenced by western culture, which places more emphasis on independence and autonomy.

Respondents who were boys or from advanced grades or had parents with a poor marital relationship were more likely to experience psychological control parenting. A previous study found that conflicts between parents will reduce a parent’s monitoring of their children and enhance overall psychological control [40]. When conflicts between parents happen, in order to transfer the conflict, couples will expect too much from their children, especially from boys, or will over-accuse and over-discipline their children, even transferring their own anger onto them or uniting the children against either the husband or wife. In these circumstances, children become a tool in the relationship for use between husband and wife. This makes the children feel that their parent’s love is not real or that they were controlled excessively in these families with frequent conflicts [41]. Children growing up in this environment are often resistant to discipline, which may, in turn, lead to more psychological control by their parents.

## 5. Limitations

Although this study has filled the research gap for parenting styles held over secondary vocational school students in China by measuring both paternal and maternal parenting practices concurrently, limitations should also be considered. Firstly, this study was conducted among secondary vocational school students in Shanghai and Shaanxi province. The findings might not be generalizable to other areas or other types of students. Secondly, our sample only covered students from three grades, therefore we fail to understand the parenting styles held over students from different age groups. Further studies are needed to explore the parenting styles held over children within different age groups and characteristics. Longitudinal data are also necessary to help us understand the development track of parenting styles during the periods from infant to emerging adulthood.

## 6. Conclusions

This paper examined the naturally formed parenting styles that are held over secondary vocational students in China via LPA. Five subtypes of parenting styles were identified within the study sample: monitoring parenting, free-range parenting, psychological control parenting, tiger parenting, and authoritative parenting, with authoritative parenting found to be the least common style. The subtypes identified in this study enriched our understanding of the parenting styles of Chinese parents and also contributed to the current knowledge on parenting styles in different cultural contexts. Findings from this study suggest that the parenting styles of many secondary vocational school students’ parents were not conducive to the healthy growth of their children, especially parents with a low economic level or a poor marital relationship. It is necessary to design interventions for the parents of secondary vocational school students to promote the authoritative parenting style. In addition, we should be concerned with the mental health of students experiencing disharmonious parental relations and provide support and help for them.

## Figures and Tables

**Figure 1 ijerph-19-07497-f001:**
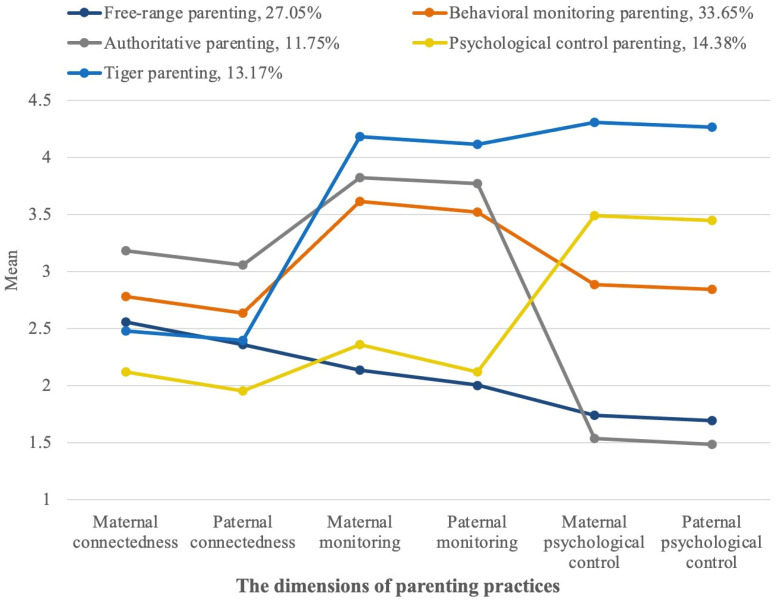
Five parenting practice profiles from the best-fitting five-class pattern.

**Table 1 ijerph-19-07497-t001:** Model fit statistics for latent profile analysis on parenting (*n* = 2392).

K	AIC	BIC	aBIC	Entropy	LMR	BLRT	Smallest Class
1	39,107.72	39,177.08	39,138.96	-	-	-	-
2	36,470.64	36,580.46	36,520.09	0.799	<0.001	<0.001	46.57%
3	35,223.38	35,373.65	35,291.05	0.843	<0.001	<0.001	28.39%
4	34,206.40	34,397.14	34,292.29	0.811	0.004	<0.001	17.34%
5	33,498.97	33,730.17	33,603.08	0.844	0.001	<0.001	11.75%
6	32,921.42	33,193.75	33,043.75	0.840	0.042	<0.001	6.30%

**Table 2 ijerph-19-07497-t002:** Mean comparisons across five latent profiles (Mean ± SD).

	Free-Range	Monitoring	Authoritative	Psychological Control	Tiger	F
**Maternal**						
Connectedness	2.555 ± 0.726	2.783 ± 0.695	3.180 ± 0.695	2.118 ± 0.664	2.479 ± 0.854	99.46 ***
Monitoring	2.136 ± 0.606	3.615 ± 0.604	3.822 ± 0.571	2.359 ± 0.658	4.179 ± 0.576	1033.79 ***
Psychological control	1.738 ± 0.540	2.886 ± 0.454	1.539 ± 0.407	3.487 ± 0.698	4.307 ± 0.542	1837.91 ***
**Paternal**						
Connectedness	2.357 ± 0.731	2.636 ± 0.694	3.056 ± 0.690	1.953 ± 0.0.618	2.398 ± 0.880	105.89 ***
Monitoring	2.003 ± 0.573	3.522 ± 0.635	3.769 ± 0.577	2.118 ± 0.584	4.112 ± 0.636	1151.31 ***
Psychological control	1.694 ± 0.534	2.842 ± 0.471	1.488 ± 0.405	3.447 ± 0.731	4.266 ± 0.578	1748.79 ***

***: *p* < 0.001.

**Table 3 ijerph-19-07497-t003:** Logistic regression on the associations between parenting substyles and adolescents’ outcomes [*OR* (95%*CI*)].

*OR* (95%*CI*)		Authoritative	Free-Range	Monitoring	Psychological Control	Tiger
Problem behaviors						
Attempt smoking *	Yes vs. no	1.00	2.33 (1.44–3.75)	2.03 (1.27–3.25)	3.02 (1.82–5.02)	2.14 (1.26–3.61)
Attempt drinking *	Yes vs. no	1.00	1.63 (1.22–2.18)	1.29 (0.97–1.70)	1.81 (1.29–2.53)	1.16 (0.83–1.62)
experience of sexual intercourse *	Yes vs. no	1.00	1.28 (0.63–2.59)	0.79 (0.38–2.80)	1.28 (0.59–2.80)	1.37 (0.62–3.02)
Academic achievement *	Average vs. Good	1.00	1.13 (0.84–1.54)	1.36 (1.01–1.82)	1.40 (0.98–2.00)	1.18 (0.83–1.68)
	Poor vs. Good	1.00	1.47 (0.91–2.35)	1.24 (0.77–1.98)	2.30 (1.38–3.85)	1.51 (0.89–2.56)
Mental health						
Depressive symptom *	Yes vs. no	1.00	1.56 (0.97–2.51)	1.58 (0.99–2.50)	3.57 (2.19–5.82)	3.03 (1.85–4.97)
Anxiety symptom *	Yes vs. no	1.00	1.56 (0.90–2.71)	1.84 (1.09–3.13)	4.11 (2.36–7.16)	4.06 (2.33–7.08)
Self-injuring attempt *	Yes vs. no	1.00	1.31 (0.66–2.58)	1.51 (0.78–2.89)	3.55 (1.81–6.96)	3.67 (1.87–7.20)
Suicide attempt *	Yes vs. no	1.00	2.01 (0.82–4.96)	2.38 (0.99–5.70)	5.12 (2.09–12.54)	5.63 (2.31–13.71)

* Covariates: gender, grade, area, one child, family economic level, and marital relationship.

**Table 4 ijerph-19-07497-t004:** Percentage distribution of respondents and percentage of parenting styles by variable (*n* = 2392).

Variables	*n* (%)	Free-Range (%)	Monitoring (%)	Authoritative (%)	Psychological Control (%)	Tiger (%)	χ^2^	*p* Value
Gender							14.4002	0.006
Boy	1121 (46.86)	28.19	32.47	9.63	16.06	13.65		
Girl	1271 (53.14)	26.04	34.70	13.61	12.90	12.75		
Grade							35.282	0.000
1	852 (35.62)	21.95	37.32	11.27	12.68	16.78		
2	788 (32.94)	29.06	31.35	12.18	16.12	11.29		
3	752 (31.44)	30.72	31.91	11.84	14.49	11.04		
Area							36.921	0.000
East	1216 (50.43)	31.66	29.03	11.02	14.97	13.32		
West	1176 (49.16)	22.28	38.44	12.50	13.78	13.01		
One-child							10.280	0.036
Yes	993 (41.51)	28.80	30.01	12.08	15.21	13.90		
No	1399 (58.49)	25.80	36.24	11.51	13.80	12.65		
Family economic level							41.795	0.000
High	654 (27.34)	26.30	32.87	16.36	11.93	12.54		
Average	1551 (64.84)	27.85	34.11	10.57	14.06	13.41		
Low	187 (7.82)	22.99	32.62	5.35	25.67	13.37		
Marital relationship							164.898	0.000
Good	1895 (79.22)	27.76	35.57	14.09	11.19	11.40		
Just so-so	426 (17.81)	25.59	27.93	3.29	23.47	19.72		
Bad	71 (2.97)	16.90	16.90	0.00	45.07	21.13		
Paternal education							14.478	0.271
Primary	294 (13.24)	26.53	33.67	12.93	14.29	12.59		
Junior	949 (42.75)	26.87	34.77	10.96	15.60	11.80		
Senior	570 (25.68)	28.95	33.86	11.40	12.11	13.68		
College	407 (18.33)	21.87	35.87	13.76	12.78	15.72		
Maternal education							16.958	0.151
Primary	529 (23.86)	25.33	34.22	12.48	16.26	11.72		
Junior	846 (38.16)	26.60	35.11	10.76	15.37	12.17		
Senior	481 (21.70)	29.31	34.51	11.02	11.02	14.14		
college	361 (16.28)	24.65	32.69	14.13	12.19	16.34		

**Table 5 ijerph-19-07497-t005:** Covariate predictors of latent class membership (compared with class 2 monitoring parenting).

RRR (95%CI)		1 vs. 2	3 vs. 2	4 vs. 2	5 vs. 2
Gender	Girl vs. Boy	0.80 (0.64–1.00)	1.36 (1.01–1.83)	0.64 (0.48–0.85)	0.86 (0.64–1.13)
Grade	2 vs. 1	1.65 (1.25–2.16)	1.41 (1.00–2.00)	1.61 (1.16–2.25)	0.81 (0.58–1.13)
	3 vs. 1	1.64 (1.25–2.16)	1.27 (0.89–1.81)	1.40 (0.99–1.98)	0.73 (0.52–1.03)
Area	West vs. East	0.46 (0.34–0.62)	1.16 (0.77–1.74)	0.64 (0.44–0.92)	0.84 (0.58–1.23)
One-child	No vs. Yes	1.02 (0.77–1.36)	0.74 (0.51–1.09)	0.86 (0.60–1.23)	0.96 (0.67–1.37)
Family economic level	Average vs. High	0.91 (0.70–1.19)	0.63 (0.46–0.86)	0.99 (0.70–1.40)	1.05 (0.75–1.46)
	Low vs. High	0.80 (0.49–1.31)	0.42 (0.20–0.88)	1.39 (0.81–2.38)	1.12 (0.62–2.00)
Marital relationship	Just so-so vs. Good	1.20 (0.88–1.63)	0.34 (0.18–0.61)	3.01 (2.16–4.19)	2.34 (1.67–3.30)
	Bad vs. Good	1.22 (0.51–2.93)	-	8.44 (4.04–17.61)	4.57 (2.04–10.26)
Paternal education	Junior vs. Primary	0.85 (0.58–1.23)	0.81 (0.50–1.30)	1.14 (0.72–1.80)	0.90 (0.56–1.45)
	Senior vs. Primary	0.74 (0.47–1.16)	0.81 (0.45–1.46)	0.96 (0.54–1.68)	0.92 (0.52–1.64)
	college vs. Primary	0.46 (0.27–0.80)	0.68 (0.34–1.37)	0.99 (0.50–1.94)	0.91 (0.47–1.79)
Maternal education	Junior vs. Primary	0.90 (0.66–1.24)	0.85 (0.57–1.29)	0.84 (0.58–1.23)	1.01 (0.67–1.92)
	Senior vs. Primary	0.88 (0.57–1.34)	0.83 (0.47–1.46)	0.56 (0.33–0.94)	1.13 (0.66–1.92)
	college vs. Primary	0.90 (0.53–1.50)	1.13 (0.58–2.19)	0.63 (0.33–1.20)	1.34 (0.71–2.52)

## Data Availability

Any data requirements can be obtained by contacting the corresponding author.

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
