# Peer review of "Parenting Practice Profiling and Its Associated Factors among Secondary Vocational School Students in China"

_ijerph, 2022, doi:10.3390/ijerph19127497_

Round 1
Reviewer 1 Report
Thank you for your invitation to review the manuscript. Overall the article is good, with a good number of sample subjects and several variables studied. The methodology is correct and the results are thoroughly studied, but in my opinion the contribution of novel concepts to the scientific community is not enough. In the following, I suggest some comments to improve your manuscript:
- In lines 65-67 you make a statement that you should clarify further. You do not understand why these types of education are not worthwhile and why parents are more active than Western parents.
- In the material and methods section they should add the informed consent of the participants to be included in the study.
- They should further explain the statement on line 126.
- The conclusions of the study show hardly any data not known so far, so I urge them to extend their study in sample, in years of schooling or even to carry out a multicentre study.
- You should review citation 16.
Thank you for everything and I remain at your disposal. Regards.
Reviewer 2 Report
The paper deals with a very important topic about profiles of parenting styles and associated factors among secondary vocational school students in China, basing on the natural sample but not a predetermined model.
The manuscript was well written with sound methodological approach to the objective of the study and with a good discussion of the results. They should indicate the software used for statistical treatment. A set of bibliographical sources adequate to the type of study and with scientific actuality, in general.
In the Background of the abstract the authors do not refer to the framework of the study, they only place the purpose of the study.
In Materials & Methods, it is not clear the citation and reference to the original scale that measures Parent-Child connectedness nor is reference made to the authors who published it, as well as to other studies carried out.
In lines 142/143 they state that "Participants were asked these 4 questions using a 4-point scale." In lines 144/145 they state that "For comparing with other two dimensions with scores from 0 to 5, we transformed the score of this dimension into 5-point when we analysed the data." They should explain why they make this change and the theoretical rationale for this methodological option.
In table 2, the legend of the 3 asterisks is not presented.
Conclusion too synthetic for the study with the importance that it has.
Review the IJERPH rules of bibliographic citation throughout the text, namely, the reference numbers should be placed in square brackets [ ], and placed before the punctuation; for example [1], [1-3] or [1,3].
Round 2
Reviewer 1 Report
Good job with the changes you have made, you have made major changes that enhance the manuscript.
Regards
This manuscript is a resubmission of an earlier submission. The following is a list of the peer review reports and author responses from that submission.